# Virtual Training System for Unmanned Aerial Vehicle Control Teaching–Learning Processes

Ricardo J. Ruiz *, Jorge L. Saravia *, Víctor H. Andaluz * 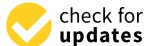 and Jorge S. Sánchez *

Departamento de Eléctrica y Electrónica, Universidad de las Fuerzas Armadas ESPE, Sangolquí 171103, Ecuador
* Correspondence: rjruiz3@espe.edu.ec (R.J.R.); jlsaravia@espe.edu.ec (J.L.S.); vhandaluz1@espe.edu.ec (V.H.A.); jssanchez@espe.edu.ec (J.S.S.); Tel.: +593-958-779-578 (V.H.A.)

**Abstract:** The present work is focused on the development of a Virtual Environment as a test system for new advanced control algorithms for an Unmanned Aerial Vehicles. The virtualized environment allows us to visualize the behavior of the UAV by including the mathematical model of it. The mathematical structure of the kinematic and dynamic models is represented in a matrix form in order to be used in different control algorithms proposals. For the dynamic model, the constants are obtained experimentally, using a DJI Matrice 600 Pro UAV. All of this is conducted with the purpose of using the virtualized environment in educational processes in which, due to the excessive cost of the materials, it is not possible to acquire physical equipment; moreover, is it desired to avoid damaging them. Finally, the stability and robustness of the proposed controllers are determined to ensure analytically the compliance with the control criteria and its correct operation.

**Keywords:** UAV; autonomous control; hexacopter; dynamic compensation; dynamic model

## 1. Introduction

With each generation for the last couple of decades, the rate at which technologic innovations are changing society has been accelerating. The way we communicate, interact, and conduct our daily work is vastly different compared to previous generations [1]. Due to this, many working areas had to adapt to this new and fast changes, being the educational sector one of the most salient. This is because education is one of the most influential factors in the progress and growth of people and societies [2,3], in which the learning and teaching process in superior education institutes is a key point for that advance. In addition to contribute with theorical knowledge, this institute offers a great reinforcement in the practical application of this concepts [4], achieving a better performance in the future generations of professionals, who can specialize in any branch of knowledge, such as medicine, construction, industrial automatization, and robotics, among others.

Robotics and automation play an important role in industries across the world. Recent technological advances have enabled robots to excel in industrial automation, gaining advantages, e.g., in improving quality and increasing production [5]. Nowadays, robots are no longer restricted only to the industrial sector; they have gradually spread to different applications in non-industrial environments and are called service robots. These can perform a variety of tasks for the entertainment or assistance in the daily life of a person [6]. Among the service robots that attract most attention in the scientific community are Unmanned Aerial Vehicles (UAVs), by virtue of the fact that they can perform completely autonomous tasks in unstructured spaces [7]. The applications of UAVs include navigation and localization [8], bridge and building inspection [9], extreme sports videography [10], autonomous detection of damage to steel surfaces by capturing panoramic images [11], and so on. Most of these areas share the same purpose to track a desired trajectory [12].

Unfortunately, the COVID-19 pandemic had an unprecedented impact on education. Classrooms were emptied and lockdowns were imposed [13]. Universities, professors,

and students were expected to adapt to the new circumstances and continue to achieve their educational goals. This forced education systems around the world to quickly switch to an emergency remote education. This mean that the institution and its users could communicate at a distance during the crisis, making greater use of Information and Communication Technologies (ICT). These are tools that transfer, process, and store information digitally [14], thus turning classrooms into virtual ones and increasing the popularity of immersive virtual environments [15].

What is sought with the virtualization of a process or scenario is to provide the user with a sense of immersion and interactivity to capture their attention. There are several ways to develop virtual environments, as detailed in [16], using different modeling languages and software, but the one that has a greater interest for its technological development is Virtual Reality (VR) in three-dimensional environments. VR is focused on stimulating a person's visual and auditory senses to replicate the experience of a real situation through a computer simulation [17]. This is a benefit when it is necessary to recreate events that may be costly or difficult to carry out in the real world. In addition, it is a great tool for training new skills remotely [18], as can be seen in the following examples. In the instrumentation area, the work presented by [19] presents a VR training system for the industrial maintenance of hydraulic pumps. For the automotive area [20] shows a low-cost VR system to simulate vehicle prototypes quickly. In the mechatronics area, ref. [21] proposes a VR environment to simulate control algorithms in simulation tasks of a wheelchair as robotic assistance and [22] also presents a unicycle robot training control in environments with hardware in the loop.

This work considers the constructivist pedagogical model, which allows students or users to contribute to their own learning process [23]. Therefore, the development of immersive and interactive virtual reality environments with users allows to simulate environments that resemble reality in different areas of knowledge, without the need for a high economic investment, or endangering the user, among other advantages [24]. Thus, nowadays, there are different strategies to capture the attention of users, for example, gamification strategies oriented to education through the development of serious games [25].

The purpose of these games is that they serve to test and explore multiple solutions to problems posed in real situations, and discover the information and knowledge that would help to intervene without fear of making mistakes [26]. Therefore, this work presents an interactive and immersive virtual training system that allows the implementation and evaluation of advanced control algorithms for the autonomous and teleoperated navigation of a UAV. The virtual system is intended to serve as a learning tool in the engineering area, specifically in the robotics area. The virtual environment is developed in the Unity3D graphics engine (Unity Software Inc., San Francisco, CA, USA). In addition, the kinematic and dynamic modeling of the UAV is incorporated with the purpose of generating greater realism in the flight animation. The mathematical models are obtained through the heuristic method and experimentally validated with the DJI Matrice 600 Pro hexacopter (DJI, Nashan District, Shenzen, China). The control scheme implements a cascade controller, which consists of a kinematic controller and another one with dynamic compensation, for which the mathematical model requires as input control signals the maneuvering velocities of the UAV. For the implementation of the advanced control algorithms the mathematical software MatLab (the MathWorks Inc., Natick, MA, USA) is considered. Therefore, a real time communication between Unity and MatLab software is considered through shared memories developed by the authors. Finally, the results obtained through the 3D virtual simulator and validated by experimental tests are presented. In addition, the usability results are presented in order to evaluate the acceptance of the developed virtual system.

The following document consists of six sections. Section 2 describes the structure of the virtual environment and the methodology that relates the teaching–learning process. Section 3 describes the UAV used, including its mathematical modeling. Section 4 explains the control scheme together with a kinematic and a dynamically compensated controller.

Section 5 presents the results obtained from experimentation and simulation, as well as the percentage of usability of the virtual training system. Finally, Section 6 contains the conclusions of the application of the virtual environment.

## 2. Methodology and Digitalization

Reality Virtual (VR) has become a broad topic of research in recent years, as an innovative solution to the problem facing higher education in times of pandemic. For the development of the work, different techniques of 3D digitization, modeling, controller design are used, including a process of experimentation.

### 2.1. Methodology

The methodology used was based on the scheme shown in Figure 1, which shows several stages of development that allow the implementation of a 3D virtual simulator.

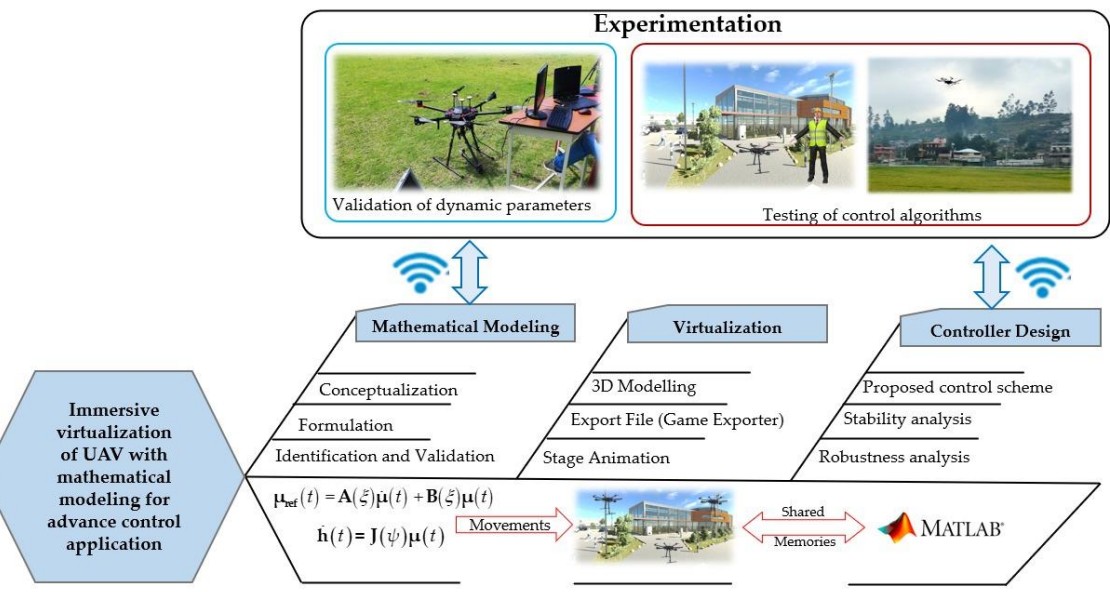

**Figure 1.** Methodology for the control and virtualization of UAVs.

The scheme proposed in Figure 1 is composed of three main stages, validated through experimental tests: (i) Mathematical Modeling is performed in order to simulate the UAV behavior in the virtual environment. Therefore, a kinematic model representing the navigation characteristics and restrictions is considered; and a dynamic model representing the dynamic behavior of the UAV-environment interaction. In addition, the identification and validation of the dynamic parameters is considered through experimental tests with the UAV Matrice 600 pro; (ii) Virtualization, both the UAV and the elements of the virtual environment are modeled using CAD tools, considering their real shapes. In addition, elements that allow simulating disturbances and different weather conditions that affect the navigation of the UAV when executing a defined task are considered. Then, by means of the 3DS Max software (Autodesk, San Francisco, CA, USA), we exported the files compatible with the Unity 3D software; (iii) Controller Design, the virtualized environment being focused for teaching–learning processes allows the testing of different proposals of advanced control algorithms, in the case of the present research the proposal is a cascade system; a controller based on the kinematic model for the tracking of the assigned trajectory considering that the robot manages to adapt perfectly to the control speeds; and a compensator based on the dynamic model, since for reasons of dynamics, reference speeds are needed to achieve the control speeds in the UAV. Then, these control speeds are communicated with the Unity 3D platform through the use of DLL libraries. Therefore, the closed control loop implemented between Unity3D and MatLab software is used at a sampling time of 100 [ms].

Finally, the tests are performed both in the real UAV Matrice 600 and in the virtualized UAV, which allows checking the operation of the proposed control algorithms and comparing with the tests in the virtual environment developed. With the purpose of validating the use of virtual environments in teaching–learning processes as test systems for new proposals of advanced control algorithms.

### 2.2. Virtual Environment

Virtual environments focused on the teaching–learning process must have real life scenes present, allowing robot–human interaction, ensuring educability. Next, the implementation of a virtual simulator that allows interaction with the virtualized hexacopter for future proposals of advanced control algorithms is detailed. In addition, the environment has elements and sounds that simulate rural and urban scenarios to increase immersion in the virtual environment. The process carried out for the Virtual Simulator was based on the scheme shown in Figure 2, where the developed sections are described.

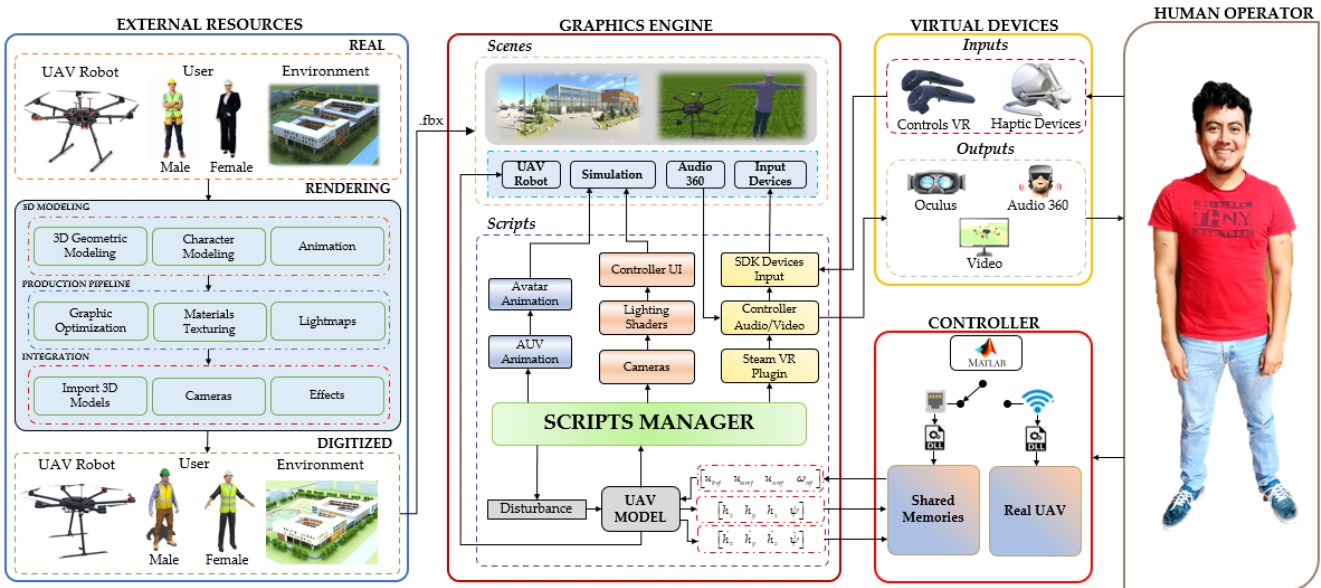

**Figure 2.** Proposed outline of the virtual simulator.

For the development of the virtual environment, the diagram in Figure 2, is made up of the following phases: (i) External Resources, includes all the elements immersed in the virtual environment, these elements can be organized mainly into three groups: (a) virtualized scenario, referred to urban, rural, and educational environments, as close to reality as possible in order to achieve learning in the environment itself for the implementation of the different advanced control algorithms for UAV trajectory tracking; (b) Virtualized UAV, the DJI Matrice 600 Pro hexacopter is digitized based on its physical characteristics and real dimensions; (c) Avatar represents the user who will use the simulator, for the digitization the anthropomorphic aspect of a human, male and female, is taken into account. The role that the avatar can play can be changed and this affects the clothing of the digital model. To perform the digitization process of these resources, CAD tools are used to model the elements, then using software such as 3DS Max and SketchUp, among others. In addition, layers are added to the elements to increase their realism, finally, the files are exported in .fbx compatible with Unity software; (ii) Graphics Engine, Unity is defined as a graphics development platform, available for Microsoft Windows, Mac OS, and Linux [27]. The development process of the virtual environment in Unity is organized in two groups:

(a) The Virtual Scenario, conformed by all the external resources digitized in .fbx format, audios, and other elements that allow the user's senses to be deceived. On the other hand, the virtual scenario is equipped with a user interface (UI), which facilitates the

user's interaction with the simulator by modifying desired tasks, physical characteristics of the environment, avatar gender selection, among others. It is also worth mentioning that this virtual scenario has implemented a real-time graphic representation system, where the evolution of each of the control errors can be observed; and (b) Programming Scripts, are one of the most relevant features when developing a 3D virtual simulator, since they allow emulating the real behavior of an Unmanned Aerial Vehicle. These movements are given by means of mathematical models both kinematic and dynamic of the UAV, considering climatic disturbances (wind speed). On the other hand, there are several scripts that have the necessary codes for the correct operation of the 3D virtual simulator. One group of these scripts allows the management of the libraries (SDK-Software Development Kit) focused on the virtual input devices, which make possible the interaction and communication between them. The remaining group of scripts manage the other components involved in the virtual scenario, such as: the UAV model, the user interface (UI), the lighting, the camera selection, the audio control, and the weather disturbances (wind speed). Together, these two groups make it possible for the virtual simulator to be interactive and immersive. (iii) Controller allows to implement advanced control algorithms capable of governing the UAV to perform trajectory tracking tasks. For this case study, the implemented scheme is based on a cascade system, with a kinematic controller and a dynamic compensation, determined through the mathematical model of the UAV. Shared memories are used as a means of communication between the Virtual Simulator developed in the Unity 3D Graphics Engine and the controller implemented in the MatLab mathematical software; on the other hand, wireless communication is used to link the controller implemented in the MatLab mathematical software, with the DJI Matrice 600 Pro UAV. Finally, (iv) Human Operator, through the virtual interface, is in charge of modifying the different parameters for the simulation, such as reference signals and disturbance data, among others, and observing the behavior of the control errors.

## 3. UAV Robot

This section describes the modelling of the UAV in order to virtualize the behavior of it for the 3D simulated scenes proposed in this work. The UAV used for this research is the DJI Matrice 600 Pro hexacopter. This work considers the kinematic modeling of the UAV, as well as the dynamic model of the robotic system.

The literature covering the mathematical modeling of UAVs is quite extensive. In recent years, much research has been based on obtaining the kinematic and dynamic models of these aerial vehicles. The purposes for which these models are used vary according to each author, but it can be agreed that in almost all cases, the mathematical model of a UAV is given as explained in the following cases [28–30]. In addition, other authors choose to represent the behavior of UAVs in a more simplified way, both for their kinematics and dynamics, some examples are [31,32]. In the following, this paper makes use of these simplified mathematical models to represent a non-linear model for the kinematics and a linear one for the dynamics.

### 3.1. Kinematic Model

The DJI Matrice 600 Pro drone is going to perform monitoring or inspection tasks, so it will track a trajectory set by the user. In this way, it will require low speeds and low value limits for the pitch angle $\theta$ and roll angle $\phi$ [33]. Therefore, the autopilot integrated in the UAV assumes that these values are negligible, although generating velocities in the front $l$ and lateral directions $m$ of the mobile reference frame $\{R_D\}$ [31]. The mathematical analysis is based on the scheme represented in Figure 3.

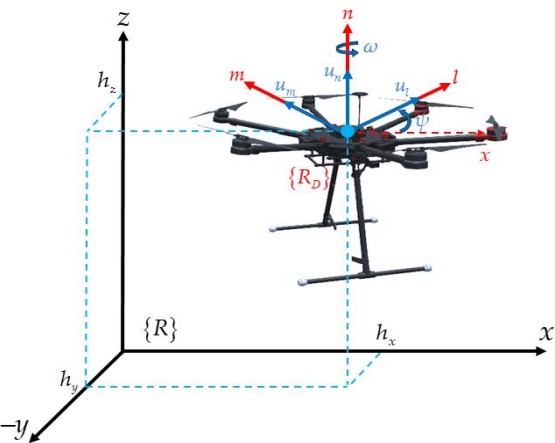

**Figure 3.** Diagram of the aerial robot.

In the mobile reference frame $\{R_D\}$ located at the center of mass of the hexacopter, the velocities are defined as follows: $u_l$ as front velocity, $u_m$ as lateral velocity, $u_n$ as elevation velocity, and the angular velocity as $\omega$, which describes the rotation of the UAV counterclockwise to the reference frame $\{R_D\}$ with respect to the axis $z$. Thus, defining the movement of the hexacopter as seen from the reference frame $\{R\}$ as follows:

$$\begin{bmatrix} \dot{h}_x \\ \dot{h}_y \\ \dot{h}_z \\ \dot{\psi} \end{bmatrix} = \begin{bmatrix} \cos\psi & -\sin\psi & 0 & 0 \\ \sin\psi & \cos\psi & 0 & 0 \\ 0 & 0 & 1 & 0 \\ 0 & 0 & 0 & 1 \end{bmatrix} \begin{bmatrix} u_l \\ u_m \\ u_n \\ \omega \end{bmatrix} \tag{1}$$

$$\dot{\mathbf{h}}(t) = \mathbf{J}(\psi)\boldsymbol{\mu}(t)$$

where $\dot{\mathbf{h}}(t) \in \mathbb{R}^m$ with $m = 4$ represents the vector of velocities of the hexacopter with respect to the reference frame $\{R\}$; $\mathbf{J}(\psi) \in \mathbb{R}^{m \times n}$ with $n = 4$ is a non-singular matrix representing the behavior of the UAV in motion, and $\boldsymbol{\mu}(t) \in \mathbb{R}^n$ represents the vector of maneuverability velocities of the UAV.

*3.2. Dynamic Model*

The dynamic model of the hexacopter is obtained by considering the UAV as a rigid body in space, which depends on the force acting on it and the torques generated by the propellers of its rotors. Thus, by using the Euler–Lagrange or Newton–Euler equations, expressions governing the translational and rotational motion of the system are obtained. However, as mentioned in [31,32], it is not necessary to develop all the dynamics of the hexacopter, simplifying it in an approximate linear model:

$$\begin{cases} \dot{u}_l = \zeta_1 u_{lref} - \zeta_2 u_l \\ \dot{u}_m = \zeta_3 u_{mref} - \zeta_4 u_m \\ \dot{u}_n = \zeta_5 u_{nref} - \zeta_6 u_n \\ \dot{\omega} = \zeta_7 \omega_{ref} - \zeta_8 \omega \end{cases} \tag{2}$$

Regrouping terms of Equation (2) in order to have a compact structure for controller design, it can be expressed as follows:

$$\begin{bmatrix} u_{lref} \\ u_{mref} \\ u_{nref} \\ \omega_{ref} \end{bmatrix} = \begin{bmatrix} \frac{1}{\zeta_1} & 0 & 0 & 0 \\ 0 & \frac{1}{\zeta_3} & 0 & 0 \\ 0 & 0 & \frac{1}{\zeta_5} & 0 \\ 0 & 0 & 0 & \frac{1}{\zeta_7} \end{bmatrix} \begin{bmatrix} \dot{u}_l \\ \dot{u}_m \\ \dot{u}_n \\ \dot{\omega} \end{bmatrix} + \begin{bmatrix} \frac{\zeta_2}{\zeta_1} & 0 & 0 & 0 \\ 0 & \frac{\zeta_4}{\zeta_3} & 0 & 0 \\ 0 & 0 & \frac{\zeta_6}{\zeta_5} & 0 \\ 0 & 0 & 0 & \frac{\zeta_8}{\zeta_7} \end{bmatrix} \begin{bmatrix} u_l \\ u_m \\ u_n \\ \omega \end{bmatrix}$$

$$\mu_{\mathbf{ref}}(t) = \mathbf{A}(\zeta)\dot{\mu}(t) + \mathbf{B}(\zeta)\mu(t) \tag{3}$$

where, $\dot{\mu}(t) = \begin{bmatrix} \dot{u}_l & \dot{u}_m & \dot{u}_n & \dot{\omega} \end{bmatrix}^T \in \mathbb{R}^m$ with $m = 4$ represents the vector of accelerations of the aerial robot with respect to the reference frame.$\{R_D\}$. $\mathbf{A}(\xi) = diag\left(\frac{1}{\xi_1}, \frac{1}{\xi_3}, \frac{1}{\xi_5}, \frac{1}{\xi_7}\right) \in \mathbb{R}^{m \times m}$ represents the inertia matrix of the aerial robot system. $\mathbf{B}(\xi) = diag\left(\frac{\xi_2}{\xi_1}, \frac{\xi_4}{\xi_3}, \frac{\xi_6}{\xi_5}, \frac{\xi_8}{\xi_7}\right) \in \mathbb{R}^{m \times m}$ represents the matrix of centripetal forces acting on the aerial robot. $\mu_{ref}(t) = \begin{bmatrix} u_{lref} & u_{mref} & u_{nref} & \omega_{ref} \end{bmatrix}^T \in \mathbb{R}^m$ represents the vector of standardized control commands of the UAV between $[-1, +1]$. Finally, we have $\zeta = \begin{bmatrix} \zeta_1 & \zeta_2 & \cdots & \zeta_l \end{bmatrix}^T \in \mathbb{R}^l$ with $l = 8$ which represents the vector containing the dynamic parameters of the aerial robot.

### 3.3. Identification and Validation

For the identification and validation process of the model, experimental tests were performed with the DJI Matrice 600 Pro hexacopter, then the data obtained were entered into the identification algorithm, which allowed finding the dynamic parameters of the model through an algorithm based on optimization and validation through the comparison of the hexacopter and the mathematical model.

This process consists of the following stages: (i) Excitation of the Hexacopter, the objective of this stage is to know the value of the output velocities before a predetermined excitation value, the difference between these signals indicates the dynamics of the hexacopter; (ii) Identification Algorithm, in this stage the dynamic parameters of the model are identified based on the data taken in the previous phase, as shown in Figure 4a. For which an algorithm based on optimization was implemented, which reduces the error resulting from comparing the values of the hexacopter with the values obtained from the mathematical model. Finally, when the error is considered negligible, the estimated dynamic parameters approximate the values of the hexacopter. Figure 4b shows the behavioral signals of the real UAV velocities $\mu_{\mathbf{r}}(t) = \begin{bmatrix} \mu_{lr} & \mu_{mr} & \mu_{nr} & \omega_{\mathbf{r}} \end{bmatrix}^T$ with respect to the excitation or reference signals $\mu_{ref}(t) = \begin{bmatrix} u_{lref} & u_{mref} & u_{nref} & \omega_{ref} \end{bmatrix}^T$ injected into the UAV.

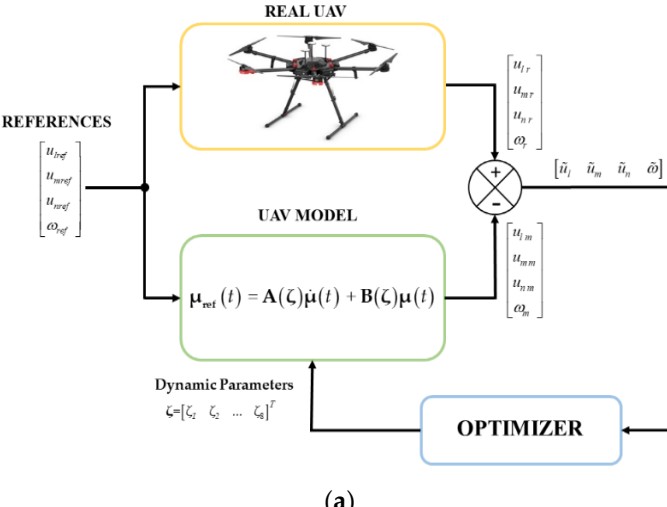

(**a**)

**Figure 4.** *Cont.*

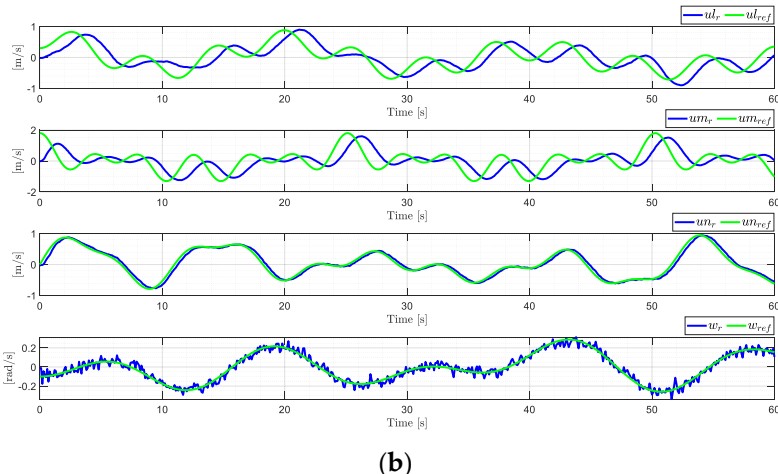

(**b**)

**Figure 4.** Identification of dynamic parameters. (**a**) Identification scheme; (**b**) Identification data signals.

(iii) Validation, the final stage allows us to evaluate whether the dynamic model with the obtained parameters represents the behavior of the hexacopter as it is seen in Figure 5a by using reference signals other than those used in the dynamic parameter identification process. The values for the dynamic parameters can be seen in Appendix A.

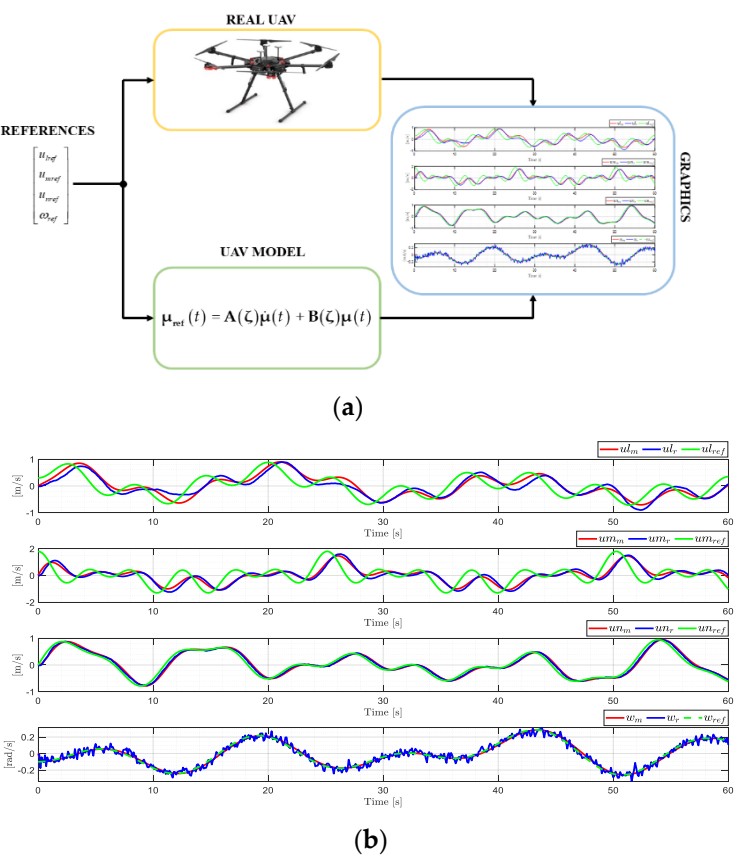

**Figure 5.** Velocities obtained from the dynamic model of the DJI Matrice 600 Pro, with the previously identified dynamic parameters, closely resemble the real UAV velocities behavior during its experimentation. (**a**) Validation scheme; (**b**) Velocity signals comparison.

## 4. Control Scheme

The proposed control scheme for the fulfillment of trajectory tracking tasks is shown in Figure 6. This scheme is based on the design in two main stages. The first stage where both

kinematic and dynamic compensation controllers are developed based on the structure of their models, respectively; on the other hand, it should be mentioned that this stage is hosted in a mathematical software in our case Matlab. Moreover, in the second stage, called virtual reality, the mathematical models are housed which allow to describe the real movements of a UAV within the 3D simulator; furthermore, this simulator is equipped with an interactive menu for the user allowing the change of the desired task as the disturbance of the process.

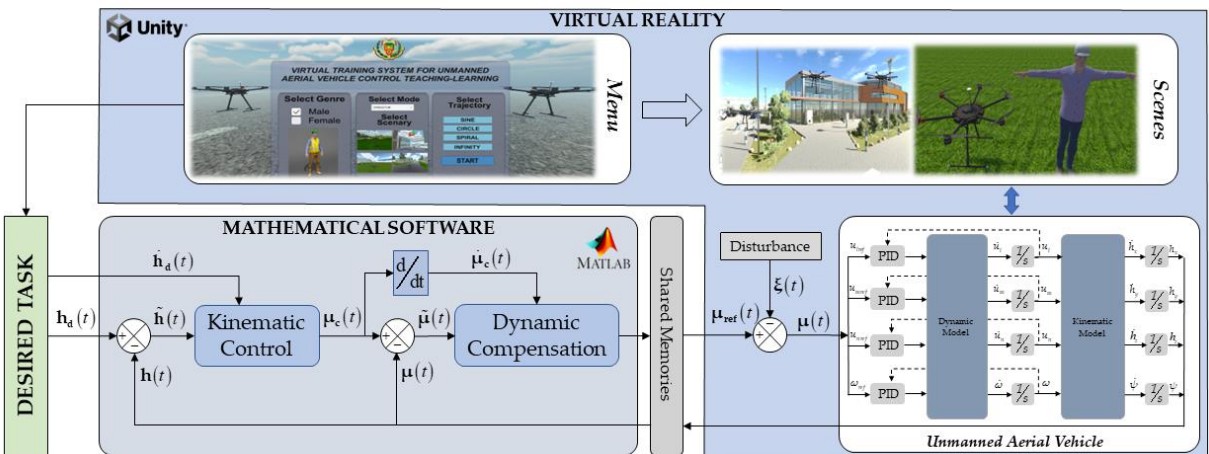

**Figure 6.** Proposed control scheme.

### 4.1. Kinematic Controller

The control errors of the UAV positions are calculated in each sampling period and are used to guide the UAV in the direction that decreases these errors. This controller is based on the kinematic model found previously in Equation (1) and is represented as follows:

$$\boldsymbol{\mu_c}(t) = \mathbf{J}^{-1}\left(\dot{\mathbf{h}}_\mathbf{d}(t) + \mathbf{K}_{\tilde{h}} tanh\left(\tilde{\mathbf{h}}(t)\right)\right) \tag{4}$$

where $\mathbf{J}^{-1}(\psi)$ represents the inverse matrix of the aerial robot kinematics $\mathbf{J}(\psi)$; $\dot{\mathbf{h}}_\mathbf{d}(t) = \begin{bmatrix} h_{dx} & h_{dy} & h_{dz} & \psi_\mathrm{d} \end{bmatrix}^\mathrm{T}$ represents the vector of desired velocities for the chosen trajectory; $\tilde{\mathbf{h}}(t) = \begin{bmatrix} \tilde{h}_x & \tilde{h}_y & \tilde{h}_z & \tilde{\psi} \end{bmatrix}^\mathrm{T}$ represents the vector of control errors; while $\mathbf{K}_{\tilde{h}}$ represents a diagonal matrix of positive gain; finally an analytical saturation $tanh(.)$ is included which limits the control error $\tilde{\mathbf{h}}(t)$.

For the kinematic controller, the behavior of the position control errors $\tilde{\mathbf{h}}(t) = \mathbf{h}_\mathbf{d}(t) - \mathbf{h}(t)$ are analyzed by considering a velocity tracking under ideal conditions, that is $\boldsymbol{\mu}(t) \equiv \boldsymbol{\mu_c}(t)$. Replacing Equation (4) in (1) we obtain the closed-loop equation $\dot{\tilde{\mathbf{h}}}(t) = -\mathbf{K}_{\tilde{h}} tanh\left(\tilde{\mathbf{h}}(t)\right)$. For the stability analysis we consider a candidate Lyapunov function defined negative $\mathbf{V}\left(\tilde{\mathbf{h}}(t)\right) = \frac{1}{2}\tilde{\mathbf{h}}^\mathbf{T}(t)\tilde{\mathbf{h}}(t) < 0$. Finally, by considering the time derivative of the candidate function $\dot{\mathbf{V}}\left(\tilde{\mathbf{h}}(t)\right) = \tilde{\mathbf{h}}^\mathbf{T}(t)\dot{\tilde{\mathbf{h}}}(t)$ and replacing it in the closed-loop equation, we obtain:

$$\dot{\mathbf{V}}\left(\tilde{\mathbf{h}}(t)\right) = -\tilde{\mathbf{h}}^T(t)\mathbf{K}_{\tilde{h}} tanh\left(\tilde{\mathbf{h}}(t)\right) < 0 \tag{5}$$

thus, guaranteeing the stability of the proposed control law, when $\mathbf{K}_{\tilde{h}} > 0$, and ensuring that $\tilde{\mathbf{h}}(t) \to 0$ it is asymptotically stable when $t \to \infty$.

### 4.2. Dynamic Compensation

The objective of the dynamic compensator is to balance the dynamics of the hexacopter in order to reduce the velocity tracking error $\tilde{\boldsymbol{\mu}}(t) = \boldsymbol{\mu_c}(t) - \boldsymbol{\mu}(t)$, which is generated by the non-perfect velocity tracking, that is $\boldsymbol{\mu}(t) \neq \boldsymbol{\mu_c}(t)$. That is why the following control law is proposed, which is based on the dynamic model (3) of the aerial robot:

$$\boldsymbol{\mu}_{ref}(t) = \mathbf{A}\left(\dot{\boldsymbol{\mu}}_{\mathbf{c}}(t) + \mathbf{K}_{\tilde{\boldsymbol{\mu}}}tanh\left(\tilde{\boldsymbol{\mu}}(t)\right)\right) + \mathbf{B}\boldsymbol{\mu}(t) \tag{6}$$

where, the control actions provided by the proposed controller are represented by $\boldsymbol{\mu}_{ref}(t) = \begin{bmatrix} u_{lref} & u_{mref} & u_{nref} & \omega_{ref} \end{bmatrix}^{\mathrm{T}}$; $\dot{\boldsymbol{\mu}}_{\mathbf{c}}(t) = \begin{bmatrix} \dot{u}_{lc} & \dot{u}_{mc} & \dot{u}_{nc} & \dot{\omega}_c \end{bmatrix}^{\mathrm{T}}$ represents the derivative of the kinematic controller velocities; $\boldsymbol{\mu}(t) = \begin{bmatrix} u_l & u_m & u_n & \omega \end{bmatrix}^{\mathrm{T}}$ represents the UAV velocities; the gain matrix to compensate for the velocity errors $\mathbf{K}_{\tilde{\boldsymbol{\mu}}}$; finally an analytical saturation $tanh(.)$ is included which limits the error $\tilde{\boldsymbol{\mu}}(t)$.

In the same way that we work with the kinematic controller, for the stability analysis of the dynamic compensator, we propose a negative candidate Lyapunov function $\mathbf{V}\left(\tilde{\boldsymbol{\mu}}(t)\right) = \frac{1}{2}\tilde{\boldsymbol{\mu}}^{\mathrm{T}}(t)\tilde{\boldsymbol{\mu}}(t) < 0$; and its time derivative $\dot{\mathbf{V}}\left(\tilde{\boldsymbol{\mu}}(t)\right) = \tilde{\boldsymbol{\mu}}^{\mathrm{T}}(t)\dot{\tilde{\boldsymbol{\mu}}}(t)$. Then we replace the control laws Equation (6) and (3) in the time derivative of the Lyapunov candidate function, we obtain:

$$\dot{\mathbf{V}}\left(\tilde{\boldsymbol{\mu}}(t)\right) = -\tilde{\boldsymbol{\mu}}^{\mathrm{T}}(t)\mathbf{K}_{\tilde{\boldsymbol{\mu}}}tanh\left(\tilde{\boldsymbol{\mu}}(t)\right) < 0 \tag{7}$$

thus, guaranteeing the stability of the proposed control law, when $\mathbf{K}_{\tilde{\boldsymbol{\mu}}} > 0$, and ensuring that $\tilde{\boldsymbol{\mu}}(t) \to 0$ it is asymptotically stable when $t \to \infty$.

### 4.3. Robustness Analysis

On the other hand, the robustness analysis is focused on the kinematic controller; specifically, the behavior of the control error in the center of mass of the hexacopter, considering that the velocity tracking is not perfect $\boldsymbol{\mu}_{ref}(t) \neq \boldsymbol{\mu}(t)$ [34]. This error in the velocity can be caused by disturbances, which is why it is defined as a Lyapunov candidate function $\mathbf{V}\left(\tilde{\mathbf{h}}\right) = \frac{1}{2}\tilde{\mathbf{h}}^{\mathrm{T}}\mathbf{h}$, with its respective time derivative $\dot{\mathbf{V}}\left(\tilde{\mathbf{h}}\right) = \tilde{\mathbf{h}}^{\mathrm{T}}\dot{\tilde{\mathbf{h}}}$. Now, considering $\boldsymbol{\mu}(t) = \boldsymbol{\mu}_{ref}(t) + \boldsymbol{\xi}(t)$ where $\boldsymbol{\xi}(t)$ represents disturbances due to climatic conditions, such as wind force, Equation (4) is substituted in (1) resulting in $\dot{\mathbf{h}}(t) = \mathbf{J}\mathbf{J}^{-1}\left(\dot{\mathbf{h}}_{\mathbf{d}}(t) + \mathbf{K}tanh\left(\tilde{\mathbf{h}}(t)\right)\right) + \mathbf{J}\boldsymbol{\xi}(t)$. In addition, considering that $\dot{\tilde{\mathbf{h}}}(t) = \dot{\mathbf{h}}_{\mathbf{d}}(t) - \dot{\mathbf{h}}(t)$, we obtain the closed loop equation expressed as:

$$\dot{\tilde{\mathbf{h}}}(t) = -\mathbf{K}_{\tilde{h}}tanh\left(\tilde{\mathbf{h}}(t)\right) - \mathbf{J}\boldsymbol{\xi}(t) \tag{8}$$

Replacing Equation (8) in the time derivative of the Lyapunov candidate function yields the expression:

$$\dot{\mathbf{V}}\left(\tilde{\mathbf{h}}\right) = -\tilde{\mathbf{h}}^{\mathbf{T}}\mathbf{K}_{\tilde{h}}tanh\left(\tilde{\mathbf{h}}(t)\right) - \dot{\tilde{\mathbf{h}}}^{\mathbf{T}}\mathbf{J}\boldsymbol{\xi}(t) \tag{9}$$

The necessary condition to fulfill that Equation (9) is negative definite is $\left| \tilde{\mathbf{h}}^{\mathbf{T}} \mathbf{K}_{\tilde{h}} tanh\left( \tilde{\mathbf{h}}(t) \right) \right| > \left| \tilde{\mathbf{h}}^{\mathbf{T}} \mathbf{J} \boldsymbol{\xi}(t) \right|$. For large values of $\tilde{\mathbf{h}}(t)$, it can be considered that $\mathbf{K}_{\tilde{h}} tanh\left( \tilde{\mathbf{h}}(t) \right) \approx \mathbf{K}_{\tilde{h}}$. With such consideration results Equation (10) as follows:

$$\| \mathbf{K}_{\tilde{h}} \| > \| \tilde{\mathbf{h}}^{\mathbf{T}} \mathbf{J} \boldsymbol{\xi}(t) \| \tag{10}$$

thus, making the errors decrease. For small values of $\tilde{\mathbf{h}}(t)$, it is considered that $\mathbf{K}_{\tilde{h}} tanh\left( \tilde{\mathbf{h}}(t) \right) \approx \mathbf{K}_{\tilde{h}} \tilde{\mathbf{h}}$, so Equation (11) can be written as:

$$\| \tilde{\mathbf{h}} \| > \frac{\| \mathbf{J} \boldsymbol{\xi}(t) \|}{\lambda_{\min}(\mathbf{K}_{\tilde{h}})} \tag{11}$$

Therefore, the error $\tilde{\mathbf{h}}(t)$ is expressed as follows:

$$\| \tilde{\mathbf{h}}(t) \| \leq \frac{\| \mathbf{J} \boldsymbol{\xi}(t) \|}{\lambda_{\min}(\mathbf{K}_{\tilde{h}})} \tag{12}$$

## 5. Experimental Analysis and Results

This section presents the virtual training system developed, as well as the implemented control scheme. This section presents the virtual training system, with its highly interactive main window that allows the modification of the different parameters immersed in the controller, as well as the configuration of the virtual environment. We also present the results obtained with the implementation of the advanced control algorithm for autonomous trajectory tracking tasks, both in the virtual training system and in the tests performed experimentally with the hexacopter.

### 5.1. Virtual Training System

For the design of the user interface within the virtual environment, the ISO 25010 standard was used as reference, which deals specifically with the usability of a software product [35]. Usability is defined as the ability of the product to be understood, learned, used, and attractive [36]. For the development of the HMI (Human Machine Interface), the double diamond model [37] was considered, for which it was needed that the virtual training system should be realistic and easy to interact with for the simulation of autonomous navigation tasks of UAVs. Likewise, the environment had to be simple enough to operate, error-free and eye-catching.

Figure 7 shows the main window of the virtual training system where the avatar's gender can be configured, allowing to choose and visualize the avatar's appearance, between male and female gender. It is also possible to select the operation mode and the scenarios, where there are two operation modes: (i) operator, which has several within the virtual simulator such as: modify the controller gains, set new trajectories, set new disturbance parameters, scenario selection, among others. (ii) observer, has several restrictions on the tasks that the operator can perform, where he can only select the trajectories set in the virtual simulator and the scenario selection. In addition, the main window allows the selection of the virtual scenario among various.

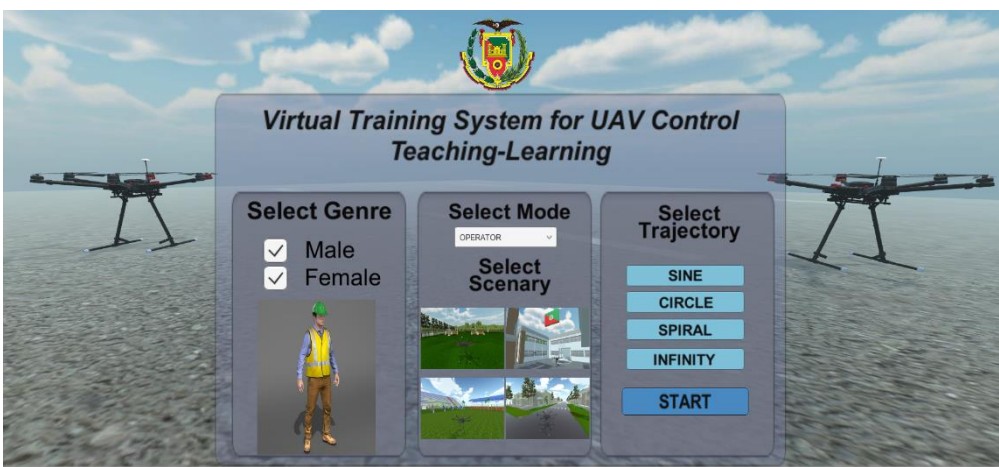

**Figure 7.** Main window of the virtual interface of the training system.

For this research, four different scenarios were developed, as shown in Figure 8. These scenarios can be classified as: (a) Park, an open place where there is abundant vegetation, such as grass, trees, and shrubs; we also find elements typical of a park, such as sidewalks and a water fountain. (b) Educational Center, there is a classroom building; it is a site with many constructions which would be obstacles for the navigation of the UAV, which has courtyards and parking lots. (c) Sports area, an open place where there is a diversity of sports venues, such as soccer stadiums, volleyball courts, tennis courts, and others, where there is a diversity of soils, such as sand, concrete, and grass, and has a little vegetation which are mostly palm trees. (d) Industrial Complex, shows a set of factories considering a moderate place in terms of space available for the execution of the task, has structures that resemble an industrial complex, and with spaces of vegetation, where trees and grass predominate.

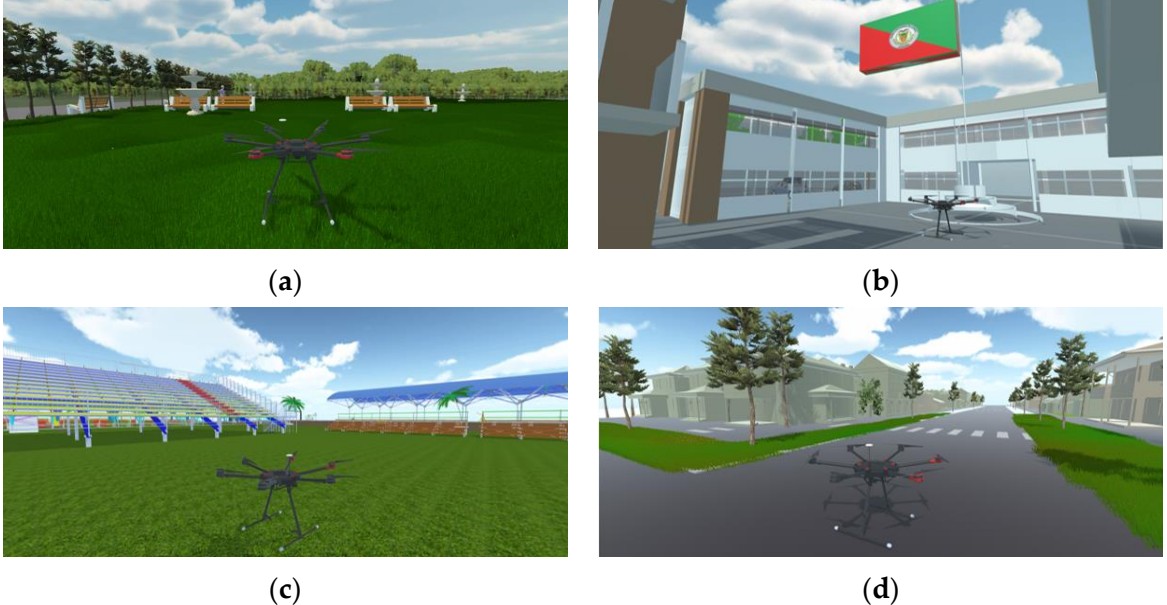

**Figure 8.** Virtual scenarios developed for the execution of trajectory tracking tasks, all related to real life. (**a**) Park Scenario; (**b**) Educational Center Scenario; (**c**) Stadium Scenario; (**d**) Neighborhood Scenario.

### 5.2. Implemented Control Scheme

This subsection presents the results of the experimentation in real life, as well as the simulation in a 3D virtual environment of the behavior of an hexacopter under the control scheme implemented for this work.

For the experimentation, the DJI Matrice 600 Pro was used as this UAV is designed for different industrial applications. It has six rotating propellers, which drive the movement of the hexacopter. It has a transmission range of 5 km and a capacity to move loads of up to 6 kg, and its 6 LiPo 6S batteries of 22.8 volts and 5700 mAh [38]. They allow it to have a flight autonomy of 38 min [39]. In addition to the standard software and hardware offered by this UAV, an Intel Nuc computer is integrated, which allows the execution of the control algorithms (see Figure 9). This modification makes it possible for control signals to be sent to the flight control board by running a Matlab script, which communicates wirelessly a remote station with the Intel board in the UAV.

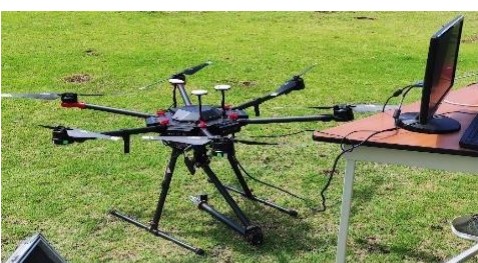

**Figure 9.** DJI Matrice 600 Pro UAV provided by the ARSI research group.

To implement the control algorithms, feedback from the positioning data obtained by the UAV's sensors, such as the D-RTK GNSS (Global Navigation Satellite System) antennas, is used. These three antennas connected to DJI's own A3 flight controller allow the UAV a vertical navigation accuracy of 0.5 m and a horizontal navigation accuracy of 1.5 m [40].

For the simulation process, the implemented control scheme is detailed in Section 2.2. The controller also runs in a Matlab script and it is in cascade form, consisting of two phases. The first one is a kinematics-based controller and the second one is a dynamic compensator, as described in Section 4. This cascade controller was based on the mathematical model of the hexacopter through the identification of the dynamic parameters based on optimization for which tests were carried out experimentally with the DJI Ma-trice 600 Pro. In this way the virtual training system becomes immersive as it best reflects the actual behavior of this UAV.

To test the performance of the advanced control scheme, an experiment is developed with the parameters described in Table 1.

**Table 1.** Desired references for the UAV.

| Coordinates | Desired Function | Initial Conditions |
|:---:|:---:|:---:|
| $h_x$ | $8\,cos(0.2t)\,[m]$ | $4[m]$ |
| $h_y$ | $7\,sin(0.4t)\,[m]$ | $1[m]$ |
| $h_z$ | $0.35\,sin(t) + 7\,[m]$ | $3[m]$ |
| $\psi$ | $tan^{-1}\left(\frac{h_y}{h_x}\right)[rad]$ | $0.5[rad]$ |

For this experiment, a displacement of the hexacopter is performed with respect to the plane *x*, *y* and *z* of the global reference frame. Figure 10 shows the trajectory of the desired task, which tends to an infinite symbol trajectory with variations in the height for the *z* axis. To analyze the performance of the implemented control scheme, the results obtained from the experimentation (see Figure 10a) are compared with those obtained from the simulation (see Figure 10b). Flight reconstruction is performed with the actual data obtained from the UAV.

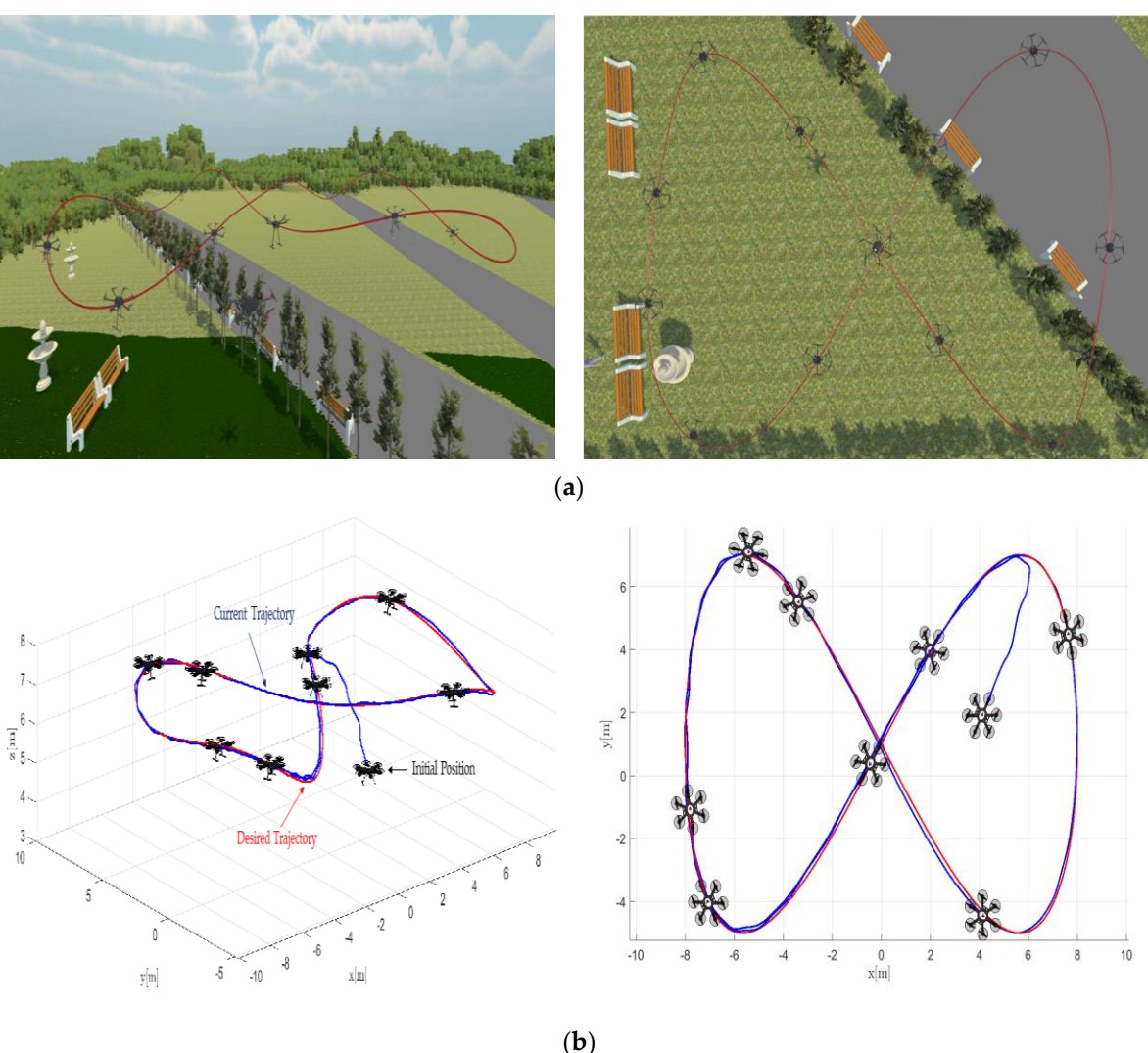

**Figure 10.** Stroboscopic movement of the UAV. (**a**) UAV stroboscopic flight based on simulated data; (**b**) UAV stroboscopic flight based on real experimental data.

In the Figure 11 show, the control errors $\tilde{\mathbf{h}}(t) = \begin{bmatrix} \tilde{h}_x & \tilde{h}_y & \tilde{h}_z & \tilde{\psi} \end{bmatrix}^{\mathrm{T}} \in \mathbb{R}^4$ tend to zero asymptotically when time tends to infinity, as well as errors in the experimentation process as in the simulation process. On the other hand, velocity errors are non-zero, as shown in Figure 12. The velocity errors are not equal to zero due to the various disturbances found in the environment where the tests were performed such as the wind force that pushes the UAV in different directions. In this way, the wind speed had a top value of 6.5 mph during the experimentation with the UAV [41]. Figure 13 shows the wind speeds during the day of experimentation. The value of the wind speed was used to accurately represent the behavior of the real UAV, as it was implemented in the simulation model.

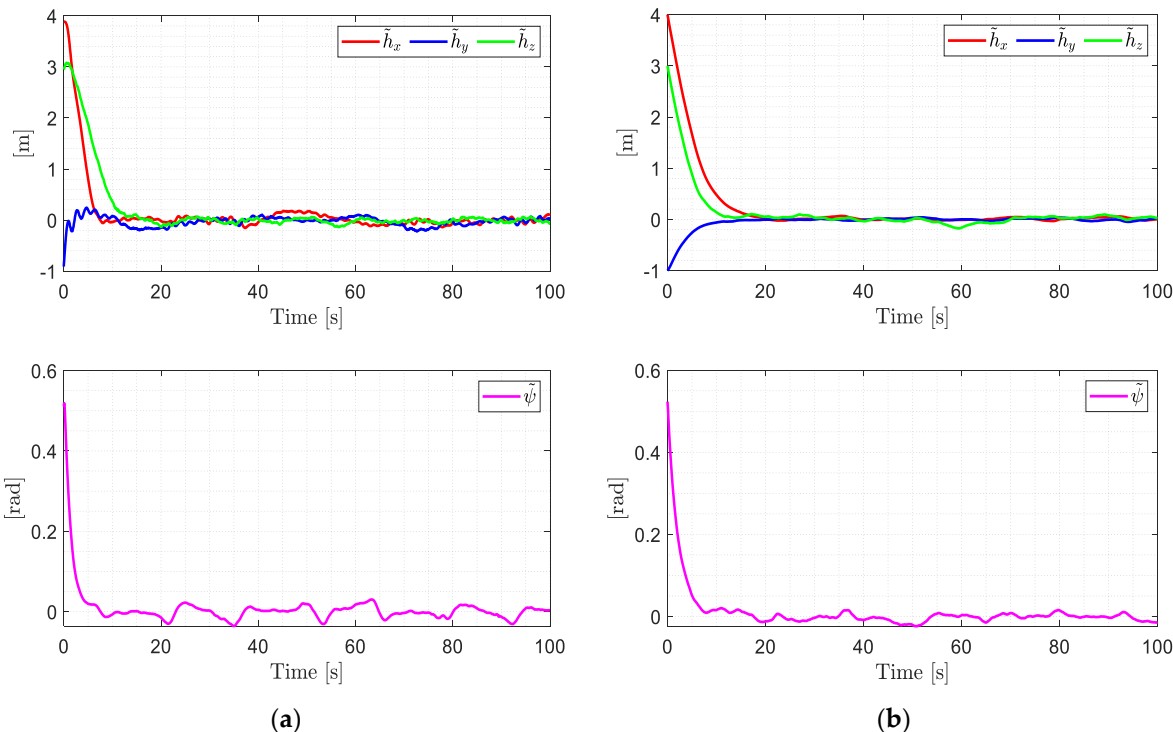

**Figure 11.** Position errors $\tilde{\mathbf{h}} = \left(\tilde{h}_x, \tilde{h}_y, \tilde{h}_z\right)$ and angle errors $\tilde{\psi}$. (**a**) Real control errors; (**b**) Simulation control errors.

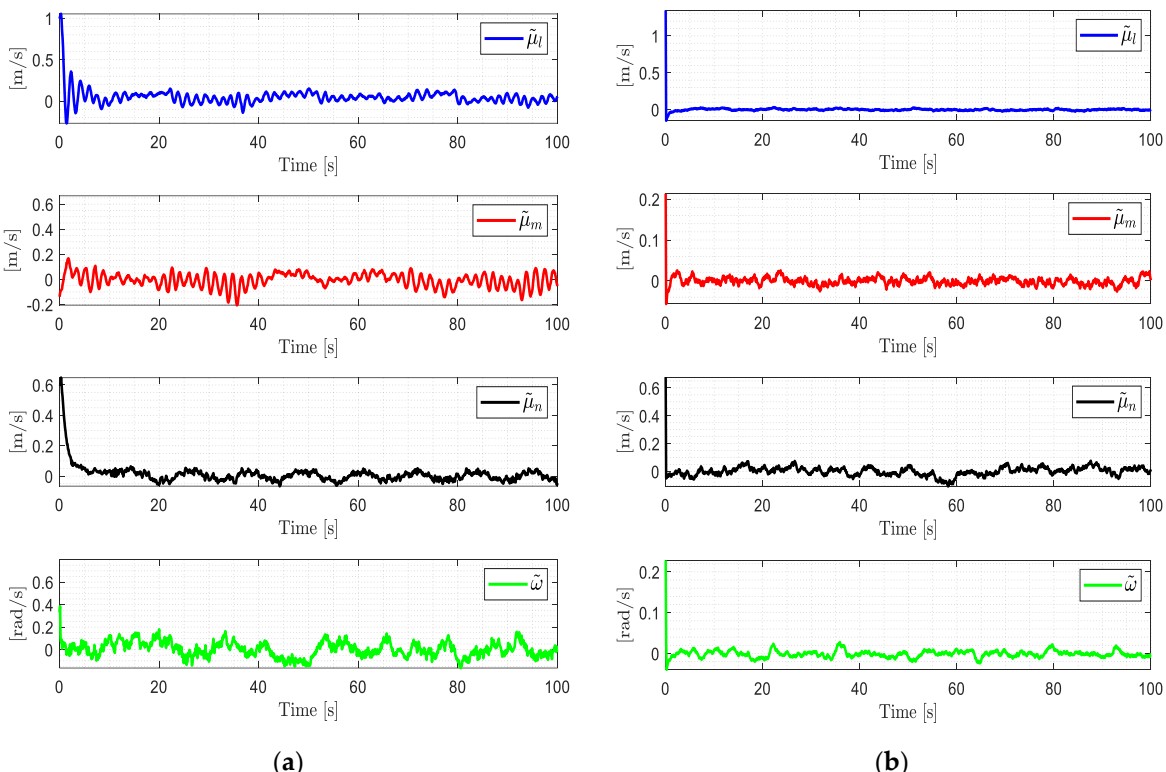

**Figure 12.** Velocity errors behavior $\tilde{\boldsymbol{\mu}} = (\tilde{u}_l, \tilde{u}_m, \tilde{u}_n, \tilde{\omega})$. (**a**) Real velocity errors; (**b**) Simulation velocity errors.

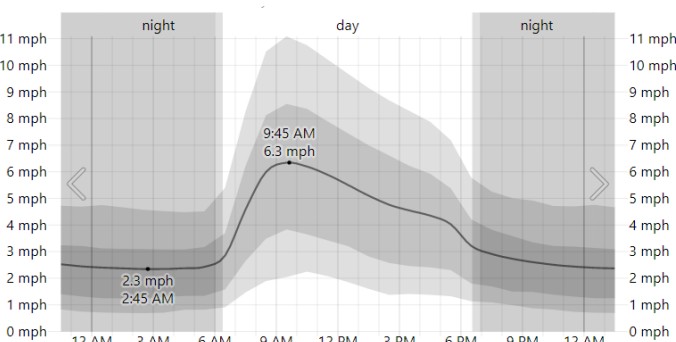

**Figure 13.** Wind speed on 2 February in Ambato. The average of mean hourly wind speeds (dark gray line), with 25th to 75th and 10th to 90th percentile bands. Civil twilight and night are indicated by shaded overlays.

Figure 14 shows the control actions applied to the hexacopter during the experiment.

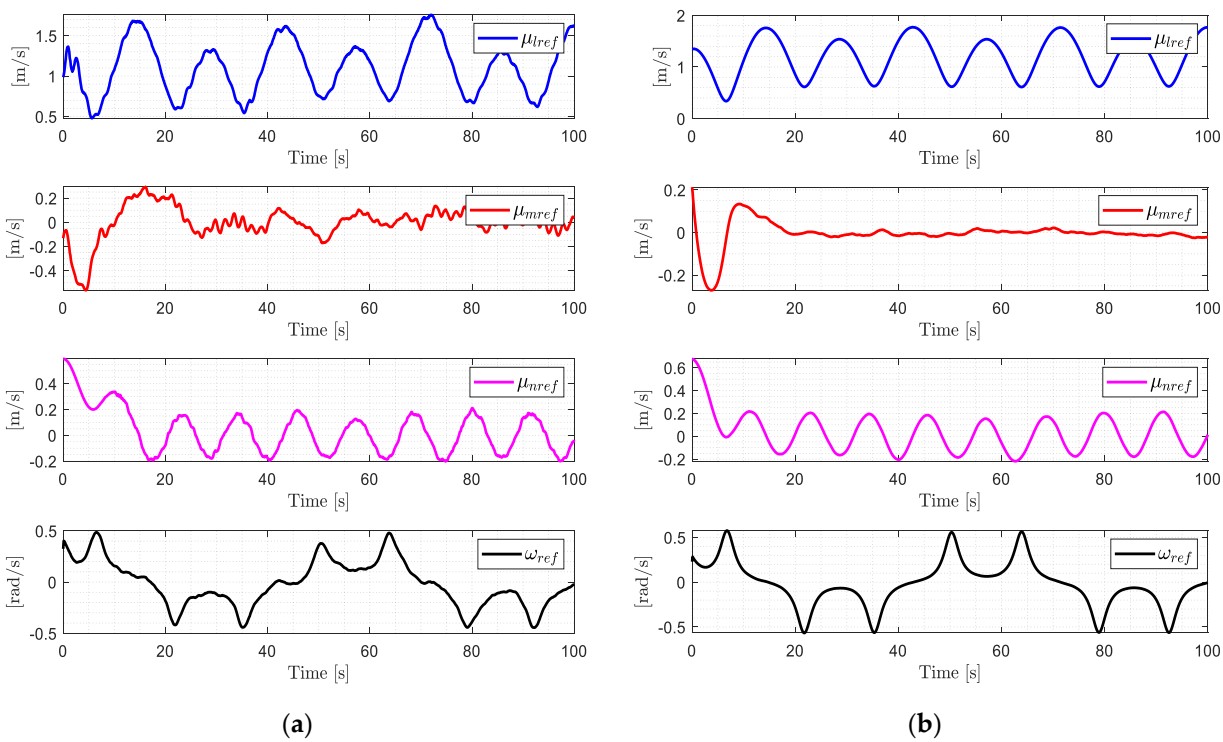

(**a**)　　　　　　　　　　　　　　(**b**)

**Figure 14.** Evolution of UAV control inputs $\mu_{ref} = \left( u_{lref}, u_{mref}, u_{nref}, \omega_{ref} \right)$. (**a**) Real input velocities; (**b**) Simulation input velocities.

Finally, it can be evidenced in the results presented that the signals: control errors, speed errors and control actions of both the experimental and validation processes have a similar behavior under the same case study, differentiating one from the other by the action of climatic conditions in the experimental process; therefore, the virtual system is a suitable environment for the implementation of various control algorithms.

### 5.3. Usability

To measure the degree of usability of the developed application, we used the System Usability Scale (SUS), which serves as a fast and reliable tool for measuring the usability attitude of a system [42]. It is a survey that gives positive results with a small sample size; in this case, we count with the help of a group of 20 people with knowledge in the area of robotics. Before the experiments, all participants were trained to navigate in VR environ-

ments. In the training, no autonomous control tasks were considered for UAV trajectory tracking. After finishing the experiments, the group completed a usability test to measure the level of acceptance of the system's features. The total average SUS score obtained was 85,375%, which indicates a good degree of usability for our virtual environment.

## 6. Conclusions

The implementation of a virtual training system for unmanned aerial vehicle control teaching–learning processes has demonstrated its capacity to simulate a scenario similar to reality. This allows future research around this application to develop a diversity of advanced controllers, observing their behavior through the evolution of control errors in diverse urban and rural scenarios. The mathematical models of kinematics and dynamics have allowed corroborating the performance of the virtual training system considering the dynamics of the UAV. This was achieved through the dynamic parameters obtained with the identification process based on optimization, for which tests were carried out experimentally with the DJI Matrice 600 Pro. The controller implemented makes possible the correction of external disturbances produced by air currents, which determines that the proposed controller is stable and robust, both in the virtual training system and in the tests carried out experimentally with the hexacopter.

**Author Contributions:** Conceptualization V.H.A., R.J.R. and J.L.S.; methodology V.H.A., R.J.R. and J.L.S.; software J.S.S., R.J.R. and J.L.S.; validation V.H.A., R.J.R. and J.L.S.; formal analysis J.S.S. and V.H.A.; investigation V.H.A., R.J.R. and J.L.S.; resources J.S.S., R.J.R. and J.L.S.; data curation R.J.R. and J.L.S.; writing—original draft preparation V.H.A., R.J.R. and J.L.S.; writing—review and editing V.H.A., J.S.S., R.J.R. and J.L.S.; visualization V.H.A., R.J.R. and J.L.S.; supervision V.H.A. and J.S.S.; project administration V.H.A.; funding acquisition V.H.A., R.J.R. and J.L.S. All authors have read and agreed to the published version of the manuscript.

**Funding:** This research received no external funding.

**Institutional Review Board Statement:** Not applicable.

**Informed Consent Statement:** Informed consent was obtained from all subjects involved in the study.

**Acknowledgments:** The authors would like to thank the Universidad de las Fuerzas Armadas ESPE for their contribution to innovation, especially in the research project "Advanced Control of Unmanned Aerial Vehicles", as well as the ARSI Research Group for their support in developing this work.

**Conflicts of Interest:** The authors declare no conflict of interest.

## Appendix A

Dynamic parameters of the DJI Matrice 600 Pro UAV.

$\zeta_1 = 0.8681$; $\zeta_2 = 0.6487$; $\zeta_3 = 0.8491$; $\zeta_4 = 0.6436$; $\zeta_5 = 2.5824$; $\zeta_6 = 2.5153$; $\zeta_7 = 4.2899$; $\zeta_8 = 4.3033$.

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
