# Peer review of "Virtual Training System for Unmanned Aerial Vehicle Control Teaching–Learning Processes"

_electronics, doi:10.3390/electronics11162613_

Round 1
Reviewer 1 Report
The paper is well written but emphasises mostly in the mathematical formulation although the title of the paper is about teaching leanring process.
Saying that, I would recommend authors to mention the eductional model they followed (i.e. game-based leanring, serious games, etc) and have a paragraph about it, somewhere in the begining of the paper.
Also, since in 5.3 usability, authors refer to HCI (Human-Computer Interaction) and UI Design and since SUS gives a relative high score (85,375%) it would be helpful for readers, the authors to mention the basic principles / methodoloy that they followed for the design of the user interface in order to achive this high SUS score.
Reviewer 2 Report
In this article, the authors proposed a Virtual Environment as a test system for new advanced control algorithms for an Unmanned Aerial Vehicles. The virtualized environment allows users to visualize the behavior of the UAV by including the mathematical model of it. However, there are several improvements need to be done before publication, such as:
1. In Figure 2, what is the purpose for demonstrating a user without a cloth?
2. In Figure 2, in the Controller part, real AUV or real UAV?
3. In the real world testing, how the ground truth and real trajectory obtained?
4. In the introduction part, the authors may add some related works in the other field to demonstrate the usage of the proposed methods, such as:
1) Luo, Cai, Leijian Yu, Jiaxing Yan, Zhongwei Li, Peng Ren, Xiao Bai, Erfu Yang, and Yonghong Liu. "Autonomous detection of damage to multiple steel surfaces from 360 panoramas using deep neural networks." Computer‐Aided Civil and Infrastructure Engineering 36, no. 12 (2021): 1585-1599.
2) Wang, Shubo, Jian Chen, Zichao Zhang, Guangqi Wang, Yu Tan, and Yongjun Zheng. "Construction of a virtual reality platform for UAV deep learning." In 2017 Chinese Automation Congress (CAC), pp. 3912-3916. IEEE, 2017.
Round 2
Reviewer 2 Report
No further question